# Emerging Roles of SWEET Sugar Transporters in Plant Development and Abiotic Stress Responses

**DOI:** 10.3390/cells11081303

**Published:** 2022-04-12

**Authors:** Tinku Gautam, Madhushree Dutta, Vandana Jaiswal, Gaurav Zinta, Vijay Gahlaut, Sanjay Kumar

**Affiliations:** 1Department of Genetics and Plant Breeding, Chaudhary Charan Singh University, Meerut 250004, India; tinkugoutam@gmail.com; 2Biotechnology Division, CSIR-Institute of Himalayan Bioresource Technology, Palampur 176061, India; madhushreed89@gmail.com (M.D.); vandana.jaiswal2009@gmail.com (V.J.); gzinta@gmail.com (G.Z.); sanjaykumar@ihbt.res.in (S.K.); 3Academy of Scientific and Innovative Research (AcSIR), Ghaziabad 201002, India

**Keywords:** sucrose transport, nectar secretion, phloem loading, gibberellin transport, CRISPR/Cas9

## Abstract

Sugars are the major source of energy in living organisms and play important roles in osmotic regulation, cell signaling and energy storage. SWEETs (Sugars Will Eventually be Exported Transporters) are the most recent family of sugar transporters that function as uniporters, facilitating the diffusion of sugar molecules across cell membranes. In plants, SWEETs play roles in multiple physiological processes including phloem loading, senescence, pollen nutrition, grain filling, nectar secretion, abiotic (drought, heat, cold, and salinity) and biotic stress regulation. In this review, we summarized the role of SWEET transporters in plant development and abiotic stress. The gene expression dynamics of various SWEET transporters under various abiotic stresses in different plant species are also discussed. Finally, we discuss the utilization of genome editing tools (TALENs and CRISPR/Cas9) to engineer SWEET genes that can facilitate trait improvement. Overall, recent advancements on SWEETs are highlighted, which could be used for crop trait improvement and abiotic stress tolerance.

## 1. Introduction

Photosynthetic organisms synthesize sugars during photosynthesis, a primary source of carbon and energy in cells [1]. Synthesized sugars are assimilated, transported, and distributed from source to sink tissues through the process of carbohydrate partitioning [1]. Sucrose is the main product of photosynthetic reactions, synthesized explicitly in the cytosol and transported to the sink organs [2]. Sucrose acts as a signaling molecule to control growth and differentiation [3]. Several review articles provide a detailed account of carbon partitioning, sugar metabolism, and signaling in plants [1,2,4,5,6,7,8]. Sugars are involved in various plant growth and developmental processes by acting as the source of carbon skeletons, the substrate of respiratory reactions, intermediate metabolites in biochemical reactions, storage substances, osmolyte, and signals in biotic and abiotic stresses [9,10,11,12,13,14]. The demand for sugar increases in the shoot/root apical meristem, flower buds, and seed/fruits organs [2,15,16]. Significant increases in sugar concentrations also occur under biotic and abiotic stresses such as cold, drought, phosphorus starvation, and pathogen attack [9,10,12,17]. In contrast, sugar levels decline under reduced oxygen conditions [9,13]. Additionally, sugars play a crucial role in regulating reproductive events such as pollen germination [18]. Thus, sugar metabolites form the core of the plant metabolism in response to developmental and environmental cues. 

Sugar transporters across cell membranes mediate sugar translocation. These are evolutionally conserved genes present in bacteria, fungi, archaea, and plants [5,19,20,21]. Sugar transporters are classified into the following three types in plants: monosaccharide transporters (MSTs), sucrose transporters (SUTs), and the most recent type, SWEETs (Sugars Will Eventually be Exported Transporters) [5,19]. The major facilitator superfamily (MFS) transporters contain MSTs and SUTs and are primarily involved in sugar influx into the cytosol. However, some MSTs, namely tonoplast sugar transporter (TST), and the vacuolar glucose transporter (vGTs) are involved in transporting sugars from the cytosol to vacuoles and act as H^+^/sugar antiporters [22]. Both MSTs and SUTs contain 12 transmembrane α-helices and mediate membrane transport of different sugars [23,24]. MSTs are localized in plasma membranes and membranes of cell organelles such as chloroplast, Golgi, and vacuoles [25,26,27,28,29,30,31]. 

The third type of sugar transporters is SWEET transporters containing 7-TM domains [19,32]. SWEETs play critical roles in phloem transport of sugars [33,34], pollen nutrition [35], nectar secretion [36], grain filling, and size regulation [37,38,39], floral transition [40], abiotic [17,41,42,43,44,45] and biotic stresses [46,47]. 

Here, we review the current state of knowledge of various biological functions of the SWEET family of sugar transporters. The role of SWEET sugar transporters in various developmental stages and abiotic stresses (i.e., drought, cold, heat and salt stress) is discussed. Additionally, how genome editing technologies such as TALENs and CRISPR/Cas9 are being utilized to engineer SWEET genes to improve agricultural traits and yield under stresses in plants.

## 2. Sugar Production in Plants

Sugar synthesis takes place in specialized plant-cell compartments known as chloroplast in the presence of sunlight and CO_2_. The produced triose-phosphate is directly transported to the cytosol or used for starch synthesis in the chloroplast. In the dark, starch degrades to hexose sugars (glucose or maltose) and gets exported to cytosol, where conversions of glucose take place [glucose to glucose-6 phosphate (G-6-P) and then G-6-P to fructose-6-phosphate]. These two products are used by sucrose-phosphate synthase to produce sucrose phosphate, which is then converted to sucrose by sucrose-phosphate phosphatase [48].

Sucrose is the primary form of sugar is gets transported over long distances in plants [48,49]. Sucrose is loaded into the phloem parenchyma via plasmodesmata (symplastic) or specialized membrane transporters (apoplastic). In the apoplastic mode of transportation, sucrose is initially transported out of mesophyll cells into apoplast via SWEETs and then imported into companion cells from the apoplast through membrane-localized sucrose/H^+^ symporters known as SUT1 [23,33] (For details see Figure 1). In some plants, particularly trees with higher plasmodesmata connectivity between the sieve element- companion cells complex and mesophyll cells, the sucrose transports in a concentration gradient manner and passively enters phloem [50]. The phloem loading strategies opted by plants have been discussed previously [51,52,53]. Sucrose accumulation attracts water, which generates enhanced turgidity directing the mass flow of assimilates toward sink tissues. The unloading of sucrose from phloem to sink cells takes place apoplasmically or symplasmically, followed by degradation carried out by cytoplasmic invertases (cINs) or sucrose synthase (SuSy) [54,55]. Additionally, cytosolic sucrose is taken up into vacuoles for hydrolysis mediated by vacuolar invertases (vINs) [56]. Cell wall invertases (cwINs) are involved in sucrose partitioning, plant development and cell differentiation and derive sink strength during pathogen infection [57]. The hexose produced from sucrose hydrolysis is further utilized in glycolysis and in the synthesis of sugar polymers (i.e., cellulose, fructan, and starch) [48].

## 3. SWEET Gene Family in Plants

Furthermore, SUT1 plays a significant role in the phloem loading of sugars [58]; however, the mechanism by which sucrose is released into apoplastic space from leaf cells remained elusive. A new type of transporters designated as SWEETs were identified by using fluorescence resonance energy transfer-based technology [19]. SWEETs are membrane-localized uniporters that transport sugars across cell membranes [19,33]. The first two foundation members of this family were identified in *Medicago truncatula* (*MtN3*) and *Drosophila melanogaster* (*Saliva*) in the late 1990s [59,60]. Therefore, the domain present in SWEET proteins is named as MtN3/Saliva (MtN3_slv). The SWEETs consisting of transmembrane helices (TMHs) and three TMHs make a 3-TM domain. In prokaryotes, SWEETs are known as semiSWEETs, since they contain only one unit of 3-TMHs (TMH1-3). In contrast, eukaryotic SWEET genes consist of two 3-TMHs units (TMH1-3 and TMH5-7) separated by a less conserved TMH (TM4) [5]. However, some uncommon SWEET proteins have also been reported in plants and oomycetes. For example, extraSWEET contains four-five repeats of 3-TM domains attached by two single TMHs reported in *Vitis vinifera* and wild rice, and superSWEET with more than five to eight repeats 3-TM domain was reported in oomycetes [61]. The structures of semiSWEET, SWEET, extraSWEET, and superSWEET are presented in Figure 2. The semiSWEETs were initially thought to be uncommon in eukaryotes. However, a recent analysis of 25 plant species showed that out of 411 SWEETs, 140 SWEET genes were present in partial forms, suggesting that they are pseudogenes with truncated domains [62]. The evolution of SWEET genes in prokaryotes and higher organisms is not clear yet. Gene duplication and fusion have been proposed as key driving forces to facilitate the evolution and distribution of SWEET transporters [32].

Plant genomes contain more SWEET genes (7–68) than animals, including humans, where only one SWEET gene is present. Drosophila has two SWEET genes, and *C. elegans* has seven SWEET genes. SWEET genes were first reported in Arabidopsis [19], then identified in other plant species (Figure 3). A phylogenetic analysis of SWEET genes in plants classified them into four clades. Genes in clade I and II encode proteins, which transport hexose sugars such as glucose and fructose. The Clade III contains genes, which encode proteins that show preferential transport activity for sucrose over glucose. Genes in the clade IV mainly include vacuolar transporters involved in a flux of fructose across the tonoplast [19,33]. However, clade V is found in mammals, Chlamydomonas, and *C. elegans* [19].

## 4. Role of SWEET Genes in Plant Development

SWEET genes are evolutionally conserved, playing a crucial role in various plant developmental processes, including phloem loading, nectar secretion, and reproductive organ development. Most of the experimental work elucidating the developmental roles of SWEETs have been performed in Arabidopsis and rice. This section describes the involvement of SWEET genes in various plant developmental processes. 

### 4.1. Nectar Secretion

Nectar secretion is an essential and complex process to attract pollinators, which helps in pollination and maintaining genetic diversity in flowering plants. Nectar is produced in specialized organs called nectaries located inside or outside flowers. Furthermore, *NEC1*, an *AtSWEET9* homolog, is predominantly expressed in the nectaries of Petunia hybrid and is assumed to play a role in nectar secretion [94]. A clade III member, SWEET9, is characterized as a nectary-specific sugar transporter in Arabidopsis, mustard and wild tobacco (*Nicotiana attenuata*) [36]. It functions in sucrose secretion from nectary parenchyma into apoplast, and mutation leads to a loss of nectar secretion. In other crops such as *Hevea brasiliensis*, *Medicago truncatula*, *Pisum sativum* orthologues of *AtSWEET9* exhibited male flower-specific abundant expression, indicating a similar function to nectar production [75,79].

### 4.2. Leaf Senescence

Leaf senescence is an important trait that influences plant yield and nutritional quality. Carbohydrate offloading is mediated by SWEET (Clade II and III) and SUT (SUT1 and SUT2) in the senescing leaves [37]. The *SWEET15*, also called *SAG29* (*Senescence-Associated Gene 29*), functions by remobilizing carbohydrates during senescence [41]. During senescence, *SWEET15* is upregulated and can be used as a senescence marker. Overexpression of *AtSWEET15* in Arabidopsis resulted in accelerated senescence that suggests its role in phloem loading during senescence [37]. The *SAG* gene (*SAG101*) encodes the membrane acyl hydrolase that regulates membrane hydrolysis in the early stages of senescence. The accumulation of hexose sugars (mainly glucose, fructose end galactose) in senescent leaves also leads to the speculation that clade II SWEETs may also function in carbon partitioning during senescence [41]. The *OsSWEET5* belongs to clade II and is involved in the galactose transporter in rice. The overexpression of *OsSWEET5* causes early leaf senescence, growth retardation, and change in auxin levels at the seedling stage in rice [42]. Increased clad II and III SWEET genes expression have been reported in senescing leaves of *Pisum sativum* and *Brassica rapa* [63,95]. In pear, the expression of *PbSWEET4* (clade III member), a homolog of *AtSWEET15*, is localized in the cell membrane. The expression of the *PbSWEET4* gene is potentially related to leaf development and is highly expressed in older leaves. The overexpression of *PbSWEET4* in strawberry plants resulted in a reduced sugar and chlorophyll content and accelerated leaf senescence [96]. Overall, this suggests that SWEET genes can be modulated to alter leaf senescence traits in plants. 

### 4.3. Fruit and Seed Development

Recent studies on gene expression in different plant species, including pineapple, apple, and pear, demonstrate the role of SWEETs in fruit development. For instance, in pineapple two genes, namely *AnmSWEET-5* and *11* demonstrated up-regulation at the early phases of fruit development [87]. In apple, nine SWEET genes were abundantly expressed during apple fruit development. Two genes, *MdSWEET9b* and *15a*, were associated with fruit sugar accumulation and likely to be implicated in fruit development [97]. Recently, in pear, histone acetylation-mediated regulation of SWEET genes was involved in fruit development [98]. Comparative transcriptomics of two pear varieties, ‘Nanguo’ (NG; low sucrose content) and its bud sport (BNG; high sucrose content) revealed that the *PuSWEET15* gene is induced in BNG fruit. *PuSWEET15* overexpression in NG fruit induced sucrose content, while silencing in BNG fruit reduced the sucrose level. 

SWEET genes also play an essential role during seed development. An increase in transcript levels of several SWEET genes (*ZmSWEET4c*, *6b*, *11*, *13a*, *13b*, *14b* and *15a*) was observed during seed germination in maize [99]. These genes participate in the sucrose efflux from scutellum to embryo axis. In crops, yield is determined by the allocation of sugars from leaves to seed, which is carried out by specific SWEET transporters. In Arabidopsis, three SWEET genes (*SWEET11*, *12*, and *15*) of clad III showed spatiotemporal expression during seed development and might help to transport sucrose from seed coat to the developing embryo. Triple mutant lines of *atsweet11:12:15* produce retarded embryos with reduced seed weight and low starch and lipid content, resulting in wrinkled seeds production [37]. In rice, the double knockout of *ossweet11:15* accrued starch in the pericarp, whereas caryopses did not comprise a functional endosperm (Yang et al. 2018). However, the knockout of a single gene in rice (*OsSWEET11*) and soybean (*GmSWEET15*) also produces the same phenotype, i.e., decreased sucrose concentration in the embryo resulting in seed abortion [100,101]. This demonstrates that SWEET transports sucrose from seed coat to the developing embryo and plays a vital role in seed development. However, in some cases, clade II SWEET genes which transport mainly hexose, are also reported to play an essential role during seed development. In maize *ZmSWEET4c*, a clade II SWEET participates in the transport of hexoses across the basal endosperm transfer layer. Impaired seed filling was observed in the mutants of *zmsweet4c* and its rice ortholog ossweet4, suggesting that SWEET4 enhances sugar import into the endosperm in both maize and rice [38]. In *Litchi chinensis*, the temporal and spatial expression profiling indicated the role of *LcSWEET2a* and *3b* in seed development [77].

### 4.4. Shoot Branching and Bud Outgrowth

Sugars are involved in shoot branching and bud outgrowth [102,103,104]. A SWEET gene (*CmSWEET17*) in *Chrysanthemum morifolium* displays axillary bud-specific expression after treatment with 20 mM sucrose, and the overexpression of *CmSWEET17* promotes axillary bud growth [105]. Simultaneously, the *CmSWEET17* overexpression lines revealed the induction of several auxin transporter genes [*AUXIN RESISTANT 1* (*AUX1*), *LIKE AUX1 2* (*LAX2*), *PINFORMED1* (*PIN1*), *PIN2* and *PIN4*], indicating that SWEET17 may be engaged in sucrose-mediated axillary bud outgrowth via the auxin transport pathway [105].

### 4.5. Development of Reproductive Organs

SWEET genes are expressed at different stages of pollen development. In Arabidopsis, *AtSWEET8/RPG1* (Ruptured pollen grain 1) is expressed in microsporocyte and tapetum. Pollen grains of atsweet8 mutants are aborted and sterile, suggesting its involvement in anther and pollen development [106]. Furthermore, *AtSWEET13/RPG2* partly restores the male fertility of *atsweet8* at the late reproductive stages, which is also expressed in the anther during microsporogenesis, indicating functional redundancy among SWEETs. However, the double mutant of *rpg1:rpg2* was fully sterile and was unable to restore [35]. Knockout mutants of *AtSWEET11* and *OsSWEET11/Os8N3/Xa13* also produced defective pollen grains and reduced male fertility in Arabidopsis and rice, respectively [107,108,109]. Some other SWEET genes such as *AtSWEET1*, *PwSWEET1*, and *AtSWEET5/VEX1* are expressed at different stages of pollen development, which indicates their role in pollen development [19,110]. 

## 5. Role of SWEET Genes in Abiotic Stress

To cope with different abiotic constraints, plants tightly regulate the vacuolar storage and transport of sugars. For instance, sugar accumulation occurs in vacuoles to minimize freezing stress [44,111]. Additionally, SWEET genes are responsive to various abiotic stresses, suggesting their role in abiotic stress response (Figure 4 and Table 1). We analyzed the transcriptional dynamics of SWEET genes using Genevestigator [112] in Arabidopsis and rice under drought, heat, cold and salt stresses (Figure 4). The following sections summarize and discuss the role of SWEET genes in plant abiotic stress responses.

### 5.1. Osmotic or Drought Stress

Prolonged drought increases the root to shoot ratio [1], which is affected by excess C assimilation in leaf that is transported to roots [113]. This suggests that sugar transporters may play a crucial role under drought stress conditions. Consistently, *AtSWEET4*, *AtSWEET13*, *AtSWEET14* and *AtSWEET15* were induced in Arabidopsis and *OsSWEET12*, *OsSWEET15* and *OsSWEET16* were induced in rice (Figure 4). Likewise, *AtSWEET11*, *AtSWEET12*, and *AtSUC2* transcript levels were significantly induced in leaves, while *AtSUC2* and *AtSWEET11-15* were induced in roots of water-stressed Arabidopsis [114]. An increase in the expression of sugar transporters in both leaves and roots suggests that plants have to maintain an efficient root system under stress conditions, which is ensured by allocating more C to roots. In contrast, Durand et al. [115] reported the downregulation of *AtSUC2*, *AtSWEET11*, *AtSWEET12*, *AtSWEET13*, and *AtSWEET15* and reduced sucrose transport between leaves and roots in response to poly-ethylene-glycol (PEG) treated Arabidopsis plants. This suggests that plants use distinct mechanisms to cope with drought and PEG-induced osmotic stress. 

**Table 1 cells-11-01303-t001:** List of important SWEET sugar transporters genes in plants involved in different abiotic stresses.

Abiotic Stress/Genes	Plant Species	Experimental Results	Reference
(**a**) Drought Stress			
*AtSWEET11*, *12*, and *15*	*Arabidopsis thaliana*	Up-regulated in shoot and roots under drought stress (0.4 g water g^−1^ compost)	[114]
*DsSWEET12* and *17*	*Dianthus spiculifolius*	*DsSWEET12* overexpression (OE) Arabidopsis lines showed enhanced tolerance to osmotic stress.	[116,117]
*GhSWEET5*, *20*, *49* and *55*	*Gossypium hirsutum*	Up-regulated under drought stress (20% PEG-6000; 1 h)	[65]
*MdSWEET17*	*Malus domestica*	*MdSWEET17* transgenic tomatoes showed higher drought tolerance (10% PEG-6000; 3–48 h)	[45]
*TaSWEET14g-1A* and *16a-4A*	*Triticum aestivum*	Up-regulated under drought stress (20% PEG-8000; 6 h)	[93]
*CaSWEET1-like*, and *4*	*Cicer arietinum*	Up-regulated under drought stress (70% available soil water fraction)	[118]
*GmSWEET6* and *15*	*Glycine max*	Up-regulated in drought stress (field water holding capacity of 35–40%)	[119]
*StSWEET10b*	*Solanum* *tuberosum*	Up-regulated under drought stress (reduced soil water content)	[120]
*AtSWEET17*	*Arabidopsis thaliana*	Up-regulated in roots under drought stress (−0.5 MPa osmotic potential; PEG-8000; 6 h)	[121]
*CsSWEET1a*, *2a*, *2c*, *3a*, *7a*, *7b* and *10a*	*Camellia sinensis*	Up-regulated under drought (PEG; 72 h)	[122]
*OsSWEET13* and *15*	*Oryza sativa*	Up-regulated under drought stress (20% PEG-6000)	[17]
(**b**) Heat Stress			
*BnSWEET9-2*, *10-3*, *12*, *13-2* and *14*	*Brassica napus*	Up-regulated under heat stress (40 °C; 3–24 h)	[63]
*GhSWEET4* and *10e*	*Gossypium hirsutum*	Up-regulated under heat stress (40 °C; 3–10 h)	[73]
*GhSWEET5*, *49* and *55*	*Gossypium hirsutum*	Up-regulated under heat stress (38 °C; 6 h)	[65]
*BrSWEET11*	*Brassica rapa*	Up-regulated under heat stress (38 °C; 8 h)	[66]
*TaSWEET14g-1A*, *14h-1B* and *15a-7D*	*Triticum aestivum*	Up-regulated under heat stress (42 °C; 6 h)	[93]
(**c**) Cold Stress			
*AtSWEET16* and *AtSWEET17*	*Arabidopsis thaliana*	Provides higher cold tolerance (1 week; 4 °C) by transporting glucose or fructose in the tonoplasts of leaves and roots	[123,124]
*AtSWEET11* and *12*	*Arabidopsis thaliana*	Up-regulated under cold stress (1 week; 4 °C) and affect vascular development	[125]
*AtSWEET4*	*Arabidopsis thaliana*	*AtSWEET4* OE lines have higher freezing tolerance	[126]
*MaSWEET1*, *4*, and *14*	*Musa acuminata*	Up-regulated under cold stress (4 °C; 22 h)	[43]
*CsSWEET16*	*Camellia sinensis*	Enhanced cold tolerance in *CsSWEET16* OE lines	[49]
*BoSWEET11b*, *11c*, *12b*, *16a*, and *17*	*Brassica oleracea*	Show variable expression pattern under cold treatments (4 °C; 3–48 h)	[64]
*MdSWEET16*	*Malus domestica*	Enhanced cold tolerance in *MdSWEET16* OE lines	[127]
*CsSWEET1a*, *1b*, *3b*, and *15c*	*Camellia sinensis*	Show variable expression pattern under cold treatments	[122]
(**d**) Salinity Stress			
*AtSWEET15*	*Arabidopsis thaliana*	*AtSWEET15*/*SAG29* OE plants show accelerated senescence and hypersensitivity to salinity	[41]
*DsSWEET17*	*Dianthus spiculifolius*	DsSWEET17 OE Arabidopsis lines have higher slat tolerance	[117]
*AtSWEET2*, *13*, *14*, *16*, and *17*	*Arabidopsis thaliana*	Show variable expression pattern under salt stress treatments (150 mM NaCl)	[128]
*MtSWEET1a*, *2a*, *2b*, *3c*, *7*, *9b*, and *13*	*Medicago truncatula*	Show variable expression pattern under salt stress treatments (300 mM NaCl)	[80]
*OsSWEET11 and 14*	*Oryza sativa*	Down-regulated under salt stress (150 mM NaCl)	[129]
*OsSWEET13 and 15*	*Oryza sativa*	Up-regulated under salt stress (20 mM NaCl)	[17]

Several other examples demonstrate the role of SWEETs in drought tolerance (Table 1). Arabidopsis seedlings overexpressing the *Dianthus Spiculifolius* gene *DsSWEET12* have longer roots, high fructose, and glucose with a lower sucrose content and higher tolerance to osmotic and oxidative stresses compared to wild type plants [116]. Transgenic lines of tomato overexpressing *Malus domestica* SWEET gene *MdSWEET17* showed a higher accumulation of fructose and enhanced drought tolerance [45]. It is well known that sugars act as osmoprotectants, and therefore participate in osmotic stress tolerance [130]. The total sugar content in embryos increased after PEG and NaCl treatment in sorghum, but the increase in fructose treatment was most apparent [131]. A significant reduction in the expression of *SWEET10b* under drought stress was also reported in potato [120].

In a recent study, another sugar transporter *AtSWEET17*, localized in the vacuolar membrane, was markedly upregulated during lateral root (LR) development under drought stress [121]. In another study, comparative root transcriptomics of chickpea under drought stress were carried out, and three SWEET genes (*N3* (LOC101510607), *SWEET1-like* (LOC101515250), and *SWEET4* (LOC101488443)) were reported to be upregulated in chickpea genotypes [118]. In soybean, drought stress increased leaf sucrose and soluble sugars but decreased root starch content. Consistent with this, the expression of sucrose transporters (*GmSUC2*, *GmSWEET6*, and *15*) was upregulated in the leaves and roots [119]. In rice, *OsSWEET13* and *OsSWEET15* were induced in response to drought stress. The higher expression of these two genes was due to the binding of an ABA-responsive TF (*OsbZIP72*) to their promoter sequences. This modulates sucrose transport and distribution in response to drought stress, thus maintaining sugar homeostasis in response to drought stress [17]. In wheat, 13 SWEET genes showed differential expression after PEG treatment at the seedling stage. These genes include four members of clade I (*TaSWEET2a1-6B*, *TaSWEET2a2-6D*, and *TaSWEET2b2-3A*), four members of clade III (*TaSWEET13c-6A*, *TaSWEET14h-6D*, *TaSWEET14g-1A*, and *TaSWEET15a-7D*), and five members of clade IV (*TaSWEET16c-4D*, *TaSWEET16a-4A*, *TaSWEET17a-5D*, *TaSWEET17c-5A*, and *TaSWEET17b-5B*). However, these genes are expressed in a clade-specific manner, i.e., the members of clade I show downregulation, clade III show upregulation, and clade IV showed upregulation [93]. Similarly, in tea plants, seven genes (*CsSWEET1a*, *2a*, *2c*, *3a*, *7a*, *7b* and *10a*) were induced under drought stress [122]. However, further research is required to address the contrasting gene expression patterns of SWEET sugar transporters under drought and osmotic conditions.

### 5.2. Heat Stress

Heat stress inhibits carbon fixation while respiration increases, and heat tolerance involves the maintenance of leaf sugar content [132,133]. Heat reduces sugar export from source leaves to sink. For instance, in maize, the export rate of sugars from source leaves decreased after heat stress [134]. Heat stress caused a decline in the starch content of tomato mesophyll cells, but it increased significantly at a later time point [135]. Consistently, *AtSWEET1*, *AtSWEET4*, *AtSWEET13* and *AtSWEET15* were induced, and *AtSWEET2, AtSWEET10* and *AtSWEET17* were suppressed in Arabidopsis. In rice, *OsSWEET14* and *OsSWEET16* were induced, and *OsSWEET3b*, *OsSWEET4* and *OsSWEET5* were suppressed (Figure 4).

In wheat, 22 sugar transporters were up-regulated and 19 were suppressed under heat stress [136]. In *Brassica napus*, SWEET genes (*BnSWEET9-2*, *10-3*, *12*, *13-2* and *14*) were up-regulated after heat stress [63]. In *B. rapa*, *BrSWEET1* was expressed after 2 h of heat stress, while *BrSWEET11* was expressed after 8 h of heat [66]. In cotton, *GhSWEET4*, *5*, *10e*, *49* and *55* showed induced expression under heat stress [65,73]. In wheat, 18 paralogues of nine SWEET genes (*SWEET1*, *2*, *3*, *4*, *6*, *14*, *15*, *16*, and *17*) showed a differential expression after 6 h of heat stress at the seedling stage. However, these genes expressed in a clade-specific manner, i.e., the members of clade II, III, and IV showed downregulation (except *TaSWEET15a-7D*), while the members of clade I showed both upregulation and downregulation [93]. Studies of different plant species under heat stress show that sugar levels of the source leaves decline due to decreased photosynthesis. However, functional gene studies are required to demonstrate the role of SWEET transporters in heat stress.

### 5.3. Cold Stress 

Cold stress induces sugar accumulation in plants, and several SWEET genes such as *AtSWEET15/SAG29* are up-regulated under cold stress [137]. The expression of *AtSWEET1*, *AtSWEET2b*, *AtSWEET4*, *AtSWEET13* and *AtSWEET15* was induced in Arabidopsis and *OsSWEET7c* and *OsSWEET14* were induced in rice (Figure 4). Cold stress resulted in a higher accumulation of glucose and fructose than wild-type in Arabidopsis sweet11 and *atsweet11/12* mutants, which resulted in cold stress tolerance. Enhanced tolerance observed in the double mutant may be due to the reduced number of xylem cells and smaller diameter vessels [125]. Additionally, it has been shown that the overexpression of *AtSWEET16* shows freezing tolerance [123]. *AtSWEET4* facilitates sugar transport in axial tissues during plant growth and development, and the transgenic plants overexpressing *AtSWEET4* exhibits higher freezing tolerance [126]. In tea, *CsSWEET16*, a vacuolar membrane transporter, is downregulated after cold stress. The overexpression of the *CsSWEET16* in Arabidopsis resulted in the compartmentation of sugars across the vacuole [44], while the overexpression of *AtSWEET17* reduced the fructose content in leaves by 80% under cold stress conditions [124,138]. Furthermore, *MaSWEET1*, *4* and *14* expressions were upregulated in banana (*Musa acuminata* L.) under various stresses, indicating its role in stress tolerance to multiple stresses [43]. A genome-wide study carried out in *Brassica oleracea* reported the downregulation of *BoSWEET11b*, *11c*, *12b*, *16a*, and *17* after chilling stress, possibly resulting in the accumulation of glucose and fructose and an enhanced chilling tolerance [64]. Similarly, several other genome-wide studies have been performed in different plants. They show differential expression of SWEET genes after cold stress (Table 1), suggesting that SWEET genes mediate cold-induced sugar-signaling responses [43,44,64,73,80,122,127]. The above studies indicate the role of SWEET genes in providing cold stress tolerance in plants; however, functional validation of these genes is still needed. 

### 5.4. Salinity Stress

Salinity stress affects various physiological and metabolic processes, ultimately inhibiting crop productivity [139]. There are two phases of salt stress in plants; the first phase is an osmotic phase in which leaf-growth inhibition occurs, followed by the second phase of ion toxicity in which accelerated leaf senescence occurs [140]. Instead, a phase zero is also suggested, known as the transient phase, and begins quickly after salt shock, resulting in a lower turgor pressure and growth rate [141]. Sucrose also behaves similarly to osmolyte and prevents salt stress-induced damages [142]. Additionally, *SWEET15* (clade III member) is mainly involved in the sucrose transportation. Our analysis showed that *AtSWEET1*, *AtSWEET2*, *AtSWEET4*, *AtSWEET14* and *AtSWEET15* were induced in Arabidopsis and *OsSWEET1b*, *OsSWEET7c* and *OsSWEET15* were induced in rice (Figure 4). Consistently, *AtSWEET15* is induced under osmotic stress, and *AtSWEET15* overexpression leads to accelerated senescence and hypersensitivity to salt stress [41]. The transcript level of *SWEET15* was observed as 64-fold higher than the control in phase 1 after salt stress, and for this property, the expression of this gene can be used as a marker to differentiate between phase 0 and phase 1 in Arabidopsis and maize [143]. The expression of *MtSWEET1a*, *MtSWEET2b*, *MtSWEET7*, *MtSWEET9b* and *MtSWEET13* were upregulated under salt, while *MtSWEET2a* and *MtSWEET3c* were down-regulated [80]. However, in Arabidopsis, a lower transcript level of *SWEET2*, *13*, *16*, and *17* was observed, while a higher transcript level was observed for *SWEET14* [128]. Since *SWEET2*, *16* and *17* transport glucose, fructose and/or sucrose across the tonoplast along the concentration gradient, their downregulation supports the hypothesis of reducing the cytosolic sugar towards storage in the vacuole [37,123]. The heterologous expression of *DsSWEET17*, a tonoplast sugar transporters of *Dianthus spiculifolius*, in *A. thaliana* affects sugar metabolism and tolerance to salinity, osmotic, and oxidative stresses [116]. In this study, higher fructose accumulation is observed in transgenic Arabidopsis than in the wild type, consistent with the previous study that reported a decrease in leaf fructose content in *sweet17* [138].

Salt stress induces the expression of *FLN2* (fructokinase-like protein2). The *FLN2* knockout generated by CRISPR/Cas9 was hypersensitive to salinity stress and showed the disruption of the sugar metabolism, inhibition of Rubisco activity, and downregulation of sucrose synthesis and transportation genes. In *FLN2* knockout lines, *SWEET11* and *SWEET14* were down-regulated after salt stress and indicated that FLN2 protein enhanced salinity tolerance in plants via influencing the sugar metabolism [129]. In rice, two SWEET genes (*OsSWEET13* and *15*) were also involved in the modulation of sucrose transport in response to salinity stress [17]. Several genome-wide studies using publicly available transcriptome data in different plant species, i.e., wheat, *Medicago truncatula*, banana, revealed differential expression of SWEET genes under salinity stress (Table 1) [80,144]. Altogether, these studies suggested that SWEET genes were also involved in salt stress tolerance in plants. However, further investigation into their functional roles in salt stress tolerance is required. 

## 6. Additional Roles of SWEETs Other than Sugar Transportation 

The SWEET members have also been demonstrated to transport hormones other than sugars. The two Arabidopsis SWEET genes (*AtSWEET13* and *AtSWEET14*) are involved in the transportation of various forms of gibberellins (GA) [145]. The Arabidopsis *sweet13;14* double mutant shows a phenotype related to the GA response, i.e., delayed anther dehiscence, and exogeneous GA application restored this phenotype. This experiment supports that SWEETs are involved in GA transport in plants. Similarly, in pea, the interaction between cytokinins, SWEET and cell wall invertase (CWIN) led to the formation of multiple shoots during pathogen infection [146]. These two studies depict the role of SWEET genes in the transportation of phytohormones and help us speculate the additional role of SWEET genes in the transport of phytohormones in addition to sugars. However, some additional research is required to confirm these extra transport functions and their vital relevance. 

## 7. Genetic Engineering of SWEET Genes for Crop Improvement

Genome editing has modernized biology and can facilitate the targeted modifications of genomes [147]. Zinc-Finger Nucleases (ZFNs), TAL Effector Nuclease (TALENs), and Clustered Regularly Interspaced Short Palindromic Repeats/CRISPR-associated protein-9 nuclease (CRISPR/Cas9) are the most commonly used tools. Several reports on rice and other plants (Cassava and cotton) exist, for which TALENs or CRISPR/Cas9 technologies have been used to target SWEET genes (Table 2). Most of these studies were focused on bacterial blight resistance. The bacterial transcription activator-like (TAL) effectors are involved in pathogen virulence. The TAL effectors of *Xanthomonas oryzae* (Xoo) transcriptionally activate rice disease-susceptibility (*S*) genes, including SWEET genes. Thus, genome-editing techniques could be used to enhance disease resistance by deleting effector-binding elements (EBEs) in the promoter region of S genes. For example, TALEs (AvrXa7, PthXo1, PthXo2, Tal5, and TalC) from different Xoo strains targets different SWEET genes (*OsSWEET11/12/13/14*) [148,149,150,151,152,153]. The EBEs in the *OsSWEET14* were edited using TALENs in a susceptible rice cv. Kitake and the mutated lines were found to be resistant towards AvrXa7 and PthXo3 strains [150,152]. For the functional study of those genes for which no naturally occurring TAL effectors are present, designer TAL effectors (dTALEs) can be used. For example, in rice *OsSWEET12* was induced by the infection of a Xoo strain transformed with dTALEs, hence providing susceptibility [151]. Additionally, in cassava (*Manihot esculenta*), by utilizing dTALE that complements TAL20Xam668 mutant phenotypes, it was shown that *MeSWEET10a* is the primary virulence gene that is the target of TAL20Xam668 [154].

Similarly, CRISPR/Cas9 technology has been implemented to edit rice SWEET genes [64,100,153,157,158,159,160,161,162,163] (Table 2). Jiang et al. [161] designed sgRNAs to edit the genes *OsSWEET11* and *OsSWEET14*, which are involved in resistance to bacterial blight caused by Xoo. In another study, the CRISPR/Cas9 approach was utilized to edit *OsSWEET13*, an *S-gene* of the pathotype PthXo2 in rice [160]. The broad-spectrum against bacterial blight resistance was achieved in rice by disrupting the EBEs of two S genes (*OsSWEET11* and *14)* by CRISPR/Cas9 system. Interestingly, the mutation was introduced into the rice cultivar Kitaake, containing the recessive resistance allele of *Xa25/OsSWEET13* [157,159,164]. Besides the disease-resistance dissections, CRISPR/Cas9 technology was also utilized to dissect the role of the SWEET gene during grain filling in rice. The knockout of *OsSWEET11/Os8N3* showed a decreased sucrose concentration in the mutant plants’ embryo sacs, which lead to aberrant grain filling. These results suggest that *OsSWEET11/Os8N3* is involved in sucrose transportation during the early phase of caryopsis development [100]. Future efforts should be carried out to target all EBE/S gene combinations and other important SWEET genes via TALEN or CRISPR/cas9 technology to confer broad-spectrum resistance, abiotic-stress resistance and plant development and growth-related traits in important crops such as rice, maize, and wheat. 

## 8. Conclusions and Future Prospects

This review highlights the role of SWEET sugar transporters in phloem-loading, symplastic sucrose transport during, pollen nutrition, nectar secretion, grain filling, biotic and abiotic stress regulation, and transport of GAs. The functional characterization of SWEETs under various developmental and stress conditions has been well documented in Arabidopsis. However, in crop plants, functional characterization studies of SWEET transporters have just begun. Gene editing tools such as TALENs or CRISPR/Cas9 can be crucial in this context and have been used to study SWEET gene function under biotic stresses, but less functional studies exist in the case of abiotic stress regulation (Table 2). Their characterization can lead to exciting discoveries as sugar plays a central role in crop growth, development, and yield. Furthermore, SWEETs can be key targets when engineering plants with an improved abiotic stress tolerance and yield.

## Figures and Tables

**Figure 1 cells-11-01303-f001:**
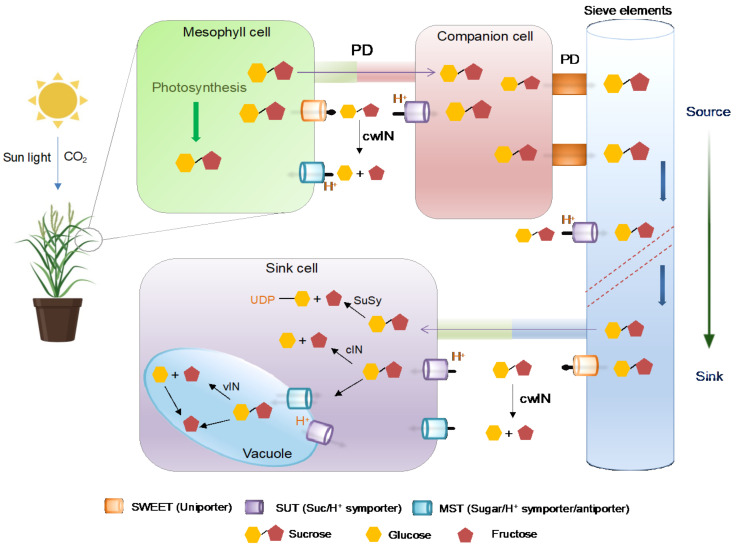
Schematic diagram showing the path of sucrose transportation from source to sink. The photosynthetically synthesized sucrose is transported out from mesophyll cells via SWEETs sugar transporters. Sucrose transporters (SUTs) accumulate sucrose in the sieve element/companion cell complex for long-distance distribution throughout the plant body. PD: plasmodesmata, cwIN: cell wall invertase, cIN: cytoplasmic invertase, SuSy: sucrose synthase, vIN: vacuole invertase, Glc: glucose, Fru: fructose.

**Figure 2 cells-11-01303-f002:**
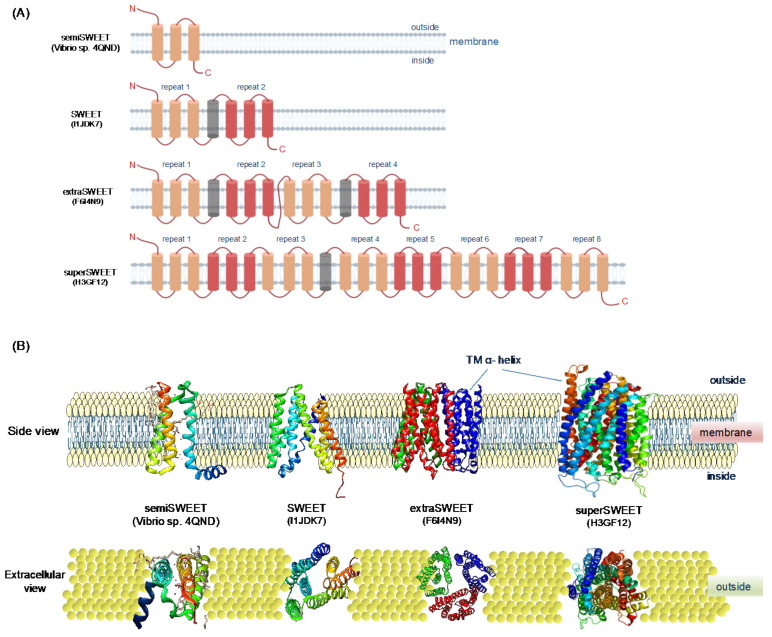
Schematic representation of two-dimensional (2D) model and 3D protein structures of four types of SWEET proteins based on 1, 2, 4 and 8, 3- transmembrane helices (TMH) domains. (**A**) Two-D models of semiSWEET, SWEET, extraSWEET and superSWEET proteins (their UniProt/PDB IDs are shown in the corresponding models). Colored boxes indicate TMHs, and loops are marked with lines. and triangles represent functional 3-TM units. (**B**) Side and extracellular view of three-D protein structures of four types of SWEET proteins. Images were prepared with PYMOL.

**Figure 3 cells-11-01303-f003:**
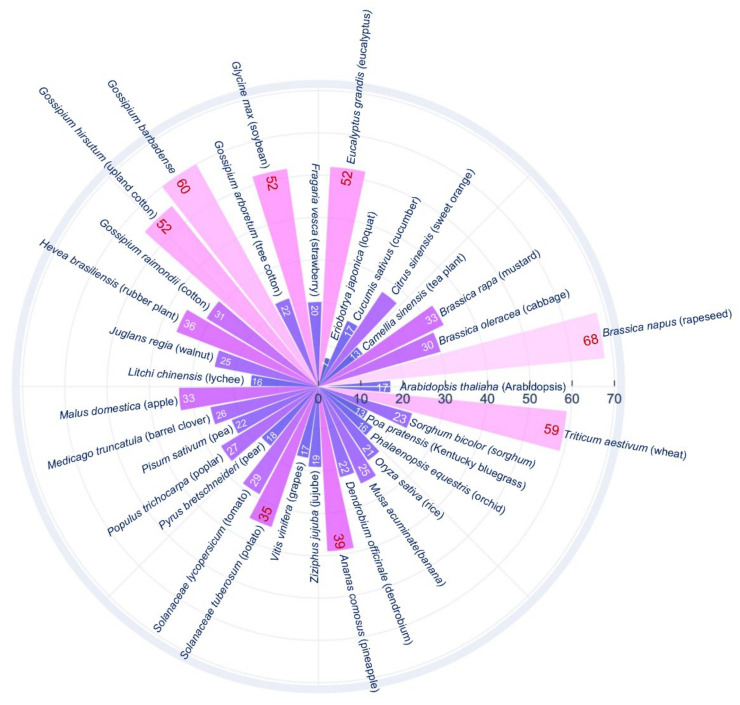
Bar plot showing the SWEET genes present in different plant species. *Arabidopsis thaliana* [19], *Brassica napus* [63], *Brassica oleracea* [64], *Brassica rapa* [65,66], *Camellia sinensis* [44], *Citrus sinensis* [67,68], *Cucumis sativus* [69], *Eriobotrya japonica* [70], *Eucalyptus grandis* [71], *Fragaria vesca* [72], *Glycine max* [62], *Gossypium arboretum* [73], *Gossypium barbadense* [73], *Gossypium hirsutum* [73,74], *Gossypium raimondii* [73], *Hevea brasiliensis* [75], *Juglans regia* [76], *Litchi chinensis* [77], *Malus domestica* [78], *Medicago truncatula* [79,80], *Pisum sativum* [79], *Populus trichocarpa* [81], *Pyrus bretschneideri* [82], *Solanaceae lycopersicum* [83], *Solanaceae tuberosum* [84], *Vitis vinifera* [85], *Ziziphus jujuba* [86], *Ananas comosus* [87], *Dendrobium officinale* [88], *Musa acuminate* [43], *Oryza sativa* [89], *Phalaenopsis equestris* [88], *Poa pratensis* [90], *Sorghum bicolor* [91], *Triticum aestivum* [92,93]. The number depicted on the bar graphs represent number of SWEET genes in the plant species.

**Figure 4 cells-11-01303-f004:**
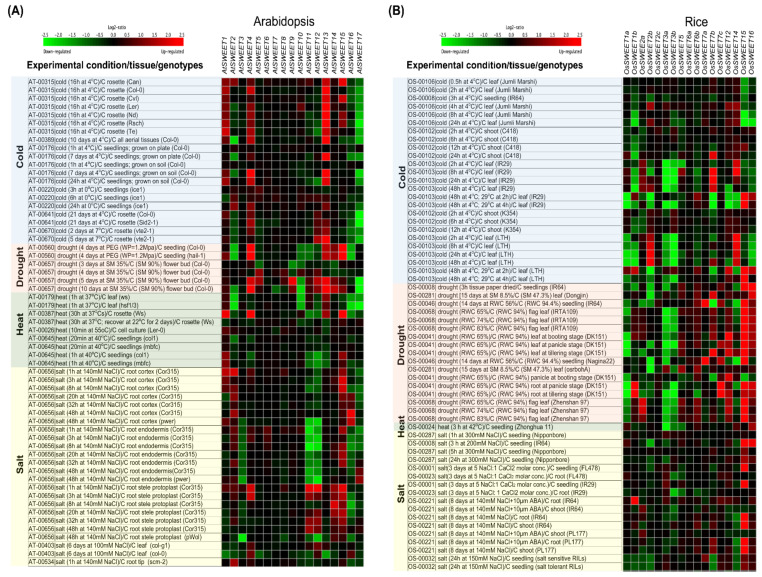
Heat maps representing SWEET gene expression patterns in (**A**) Arabidopsis and (**B**) rice under abiotic stresses (cold, drought, heat and salt). Heatmap was constructed from the data obtained from the Genevestigator database containing different experiments. [Red = up-regulation and Green = down-regulation of genes].

**Table 2 cells-11-01303-t002:** Genome editing approaches utilized to target SWEET genes in plants.

Diseases or Trait/Plant Species	Target SWEET Gene	Genome Editing Approaches Used (TAL Effectors)	Reference
(**a**) Bacterial Blight Resistant			
Rice	*OsSWEET11/Os8N3/Xa13*	TALENs (PthXO1)	[19]
Rice	*OsSWEET12*	TALENs (ArtTALs)	[155]
Rice	*OsSWEET14/Os11N3*	TALENs (AvrXa7, PthXO3)	[148,149,150,152]
Rice	*OsSWEET15*	TALENs (ArtTALs)	[151]
Cassava	*MeSWEET10a*	TALENs (TAL20Xam668)	[154]
Cotton	*GhSWEET10*	TALENs (Avrb6)	[156]
Rice	*OsSWEET11, 14*	CRISPR/Cas9	[157,158]
Rice	*OsSWEET11/Os8N3*	CRISPR/Cas9	[159]
Rice	*OsSWEET13/Os12N3/Xa25*	CRISPR/Cas9	[160]
Rice	*OsSWEET14, 11*	CRISPR/Cas9	[161]
Rice	*OsSWEET14*	CRISPR/Cas9	[153]
Cassava	*MeSWEET10a*	CRISPR/Cas9	[61]
(**b**) Grain Filling			
Rice	*OsSWEET11/Os8N3/Xa13*	CRISPR/Cas9	[100]

## Data Availability

Not applicable.

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
