# Peer review of "Emerging Roles of SWEET Sugar Transporters in Plant Development and Abiotic Stress Responses"

_cells, 2022, doi:10.3390/cells11081303_

Round 1

Reviewer 1 Report

This is a quite well-written manuscript. The authors made a comprehensive review of literature data relating to sugar transporters SWEET, and they refer both to some old papers as well as to the newest ones. In my opinion, this article may be interesting to many scientists and I recommend acceptance of it. Although I generally have no merit objections, I recommend some improvements.

The abstract should be improved. There is one sentence “Here, we provide a detailed view on the different roles of SWEET transporters in plants”. ‘Different’ means what? Instead of ‘different’, it should be written what the new roles are.

Keywords should be changed because keywords should not be the same as words already used in a title.

The citing of references is not as the journal requires. All references must be numbered [1], [2-4], etc. Also, the list of references is prepared not in accordance with the journal requirements.

Introduction. I recommend adding one reference in this section. Morkunas I, Borek S, Formela M, Ratajczak L 2012. Plant responses to sugar starvation. In: Chuan-Fa Chang (ed.) Carbohydrates - Comprehensive Studies on Glycobiology and Glycotechnology, InTech, chapter 19: 409-438, doi: http://dx.doi.org/10.5772/51569. It is the book chapter describing in detail the role of sugars in plants and in my opinion, it will be appropriate for support the information provided in the Introduction.

Section 2 and Figure 1. The abbreviations like CC or SE on page 2 are not necessary. The abbreviations like CIN, VIN, and CWI are written in a different way as in Figure 1. The writing of these abbreviations must be the same both in the text and in the figure. Instead of MC, CC, SE, and SC used in Figure 1, I recommend using the full names of particular components of the scheme directly in the scheme (instead of MC it can be simply written mesophyll cells; without abbreviation). It will improve the reading of the figure. SuSy is mistakenly written in the sink cell. And the last point: V is missing in the figure legend.

Figure 4 is difficult to read due to the small fonts. Is it possible to enlarge them?

Reviewer 2 Report

The review on SWEET transporters by Gautam et al. is well written and provides a comprehensive synthesis of the information available on these important proteins in different plants. The illustrations are clear and useful. I noted several minor errors in Fig . 2 that should be corrected.

Minor points: Line 96-97 :” The unloading of sucrose from phloem to sink cells takes apoplasmically or symplasmically”:

 please check sentence, a word is missing

Line 113: “Although SUT1 plays a significant role in the phloem loading of sugars (Riesmeier et al. 1994).”

Please check this sentence and the following one (-awkward repetitions of “however”)

“However, some uncommon types of SWEET proteins have also been reported in plants. For example, extraSWEET contains four 3-TM domains attached by two single TMHs and superSWEET with more than 5-8 repeats of 3-TM domain (Jia et al. 2017):

Please precise to what kind of plants you refer;

Figure 2 : IJDK7 or LJDK7 or 1JDK7 does not correspond to any Uniprot or PDB entries. Please correct.

4QND is the structure of a Vibrio sp. N418 protein (not a plant protein).

In Figure 2 the UniProt accession H3GF12 is a Phytophtora protein (not a plant protein) and the PDB entry 6AFW corresponds to Proton pyrophosphatase-T228D mutant: please explain the relation between supersweet protein and 6AFW protein.

Lin 212: Several recent studies on gene expression analysis studies in different plant species,: awkward repetition of “studies”; please correct.

Lines 277-278: The following sections summarize and discuss the role SWEET genes in during abiotic stresses in plants: a word is missing.

Line 290: Likewise, AtSWEET11, AtSWEET12, and AtSUC2 transcript levels were significantly induced in leaves, while AtSUC2 and AtSWEET11-15 were induced in roots of water deficit Arabidopsis (Durand et al. 2016).

Do the authors mean “deficient” ?

Line 353: showed differential expression after 6 h of HS at the seedling: please replace “HS” by heat stress

Line 366: atsweet11

Line 401: However, under normal condition, high mortality observed in root cells of transgenic plants (Seo et al. 2011). : check sentence.

Line 424: In rice, two SWEET sugar porter: do the Authors mean uniporter ? transporter ?

Line 454: please define EBEs

Line 464 : utilizing dTALE that compliments : complements ?

The authors might reference the ChloroKB website in which a metabolic map is specifically devoted to sugar transport in Arabidopsis cell (http://chlorokb.fr/metabolicMap/show/5fbd51d2bf87fb7420000031?mol=Sugar_transport )
